# The Potential and Emerging Role of Quantitative Imaging Biomarkers for Cancer Characterization

**DOI:** 10.3390/cancers14143349

**Published:** 2022-07-09

**Authors:** Hishan Tharmaseelan, Alexander Hertel, Shereen Rennebaum, Dominik Nörenberg, Verena Haselmann, Stefan O. Schoenberg, Matthias F. Froelich

**Affiliations:** 1Department of Radiology and Nuclear Medicine, University Medical Center Mannheim, Medical Faculty Mannheim of the University of Heidelberg, 68167 Mannheim, Germany; hishan.tharmaseelan@medma.uni-heidelberg.de (H.T.); alexander.hertel@umm.de (A.H.); shereen.rennebaum@umm.de (S.R.); dominik.noerenberg@umm.de (D.N.); stefan.schoenberg@umm.de (S.O.S.); 2Institute of Clinical Chemistry, University Medical Center Mannheim, Medical Faculty Mannheim of the University of Heidelberg, 68167 Mannheim, Germany; verena.haselmann@medma.uni-heidelberg.de

**Keywords:** oncologic imaging, quantitative imaging biomarkers, radiomics, precision medicine

## Abstract

**Simple Summary:**

Modern, personalized therapy approaches are increasingly changing advanced cancer into a chronic disease. Compared to imaging, novel omics methodologies in molecular biology have already achieved an individual characterization of cancerous lesions. With quantitative imaging biomarkers, analyzed by radiomics or deep learning, an imaging-based assessment of tumoral biology can be brought into clinical practice. Combining these with other non-invasive methods, e.g., liquid profiling, could allow for more individual decision making regarding therapies and applications.

**Abstract:**

Similar to the transformation towards personalized oncology treatment, emerging techniques for evaluating oncologic imaging are fostering a transition from traditional response assessment towards more comprehensive cancer characterization via imaging. This development can be seen as key to the achievement of truly personalized and optimized cancer diagnosis and treatment. This review gives a methodological introduction for clinicians interested in the potential of quantitative imaging biomarkers, treating of radiomics models, texture visualization, convolutional neural networks and automated segmentation, in particular. Based on an introduction to these methods, clinical evidence for the corresponding imaging biomarkers—(i) dignity and etiology assessment; (ii) tumoral heterogeneity; (iii) aggressiveness and response; and (iv) targeting for biopsy and therapy—is summarized. Further requirements for the clinical implementation of these imaging biomarkers and the synergistic potential of personalized molecular cancer diagnostics and liquid profiling are discussed.

## 1. Introduction

### 1.1. Achievements in Personalized Cancer Therapy

In modern medicine, cancer remains a tremendous therapeutic challenge. In 2018, it caused 1.4 million deaths and cost approximately EUR 199 billion in Europe [1]. Through better prevention, e.g., occupational safety, vaccination against oncogenic such as the human papillomavirus [2], and early detection measures such as colorectal cancer screening [3,4], the mortality of many cancers is decreasing [5]. Despite these improvements, it is still the second leading cause of global death for non-infectious diseases [6].

Modern cancer therapy primarily consists of multiple treatment modalities, including surgery, chemotherapy, radiation therapy, targeted antibodies, immunotherapy (including CAR-T-cells), and interventional radiology therapy. In general, treatment has changed significantly from mainly applying standardized surgery and chemotherapy protocols to personalized biological and immunotherapy protocols. This personalization of therapy has led to increased survival in cancer patients; for example, the anti-EGFR antibody Cetuximab used in the treatment of RAS-wildtype EGFR-expressing advanced colorectal cancer shows improved overall (hazard ratio: 0.77) and progression-free survival (hazard ratio: 0.68) compared to standard of care [7]. In terms of interventional radiology therapies, transarterial chemoembolization (TACE) and ablation therapies have enabled the targeting of specific lesions in clinical routine [8].

### 1.2. Current Status of Oncologic Imaging: Response Assessment

Since a broad range of therapies is available for solid tumors, individual characterization of therapy response is essential. The Response Evaluation Criteria in Solid Tumors (RECIST) in version 1.1 are currently widely used for response evaluation in solid tumors for clinical trials and beyond [9,10]. This very successful model was defined by the RECIST Working Group and initially intended for the standardization of imaging-guided therapy response evaluation in clinical studies.

The response of a patient is estimated by the definition of target- and non-target lesions and structured evaluation of growth patterns of tumor lesions. The criteria are defined by measurement on imaging and are widely used in clinical studies. The reproducibility of the RECIST criteria was shown to be limited in several diseases, such as prostate cancer, glioblastoma, and hepatocellular carcinoma [11]. The evaluation of response is often limited to tumor volume-shrinking cytostatic therapy. A solely size-based assessment is often not accurate in predicting survival [12], as modern non-cytostatic treatments that are used increasingly rely on the blockade of receptors to achieve their therapeutic effect, not only through tumor volume reduction. For example, inhibition of further growth may positively influence survival but may not be captured by the classic RECIST criteria [13]. As an alternative to RECIST, Choi criteria based on size and change in tumor attenuation in CT were developed to evaluate response in gastrointestinal stromal tumors that were treated with tyrosine kinase inhibitors [14]. In addition, they have been shown to be superior with respect to other oncologic entities and treatments, such as transarterial radioembolization in patients with hepatocellular carcinoma [15]. To evaluate these methods and support decision-making, more precise and objective criteria for response need to be developed.

For the improvement of the quantification of response, additional diameter-based criteria, such as early tumor shrinkage (ETS) and depth of response (DpR), have been proposed [16,17]. To address the drawbacks of diameter-based assessment, volumetric assessment of lesion size has been proposed as a more accurate alternative [18]. While this approach may help to improve the accuracy of solely lesion diameter-based measurement, the characterization of tumoral heterogeneity between lesions remains focused on changes in size. Specifically, an influence of initial lesion size and location can be noted [19]. Furthermore, an increased mean baseline CT-/Hounsfield attenuation of liver metastases could be shown to be associated with prolonged overall survival in metastatic colorectal cancer [20].

### 1.3. Current Status of Oncologic Imaging: Characterization of Lesions

Individual lesion analysis becomes progressively more feasible on the diagnostic side as modern methodologies, such as epigenetics, proteomics, transcriptomics, and genomics, provide additional information to traditional, biopsy-derived histological markers, such as grading of differentiation and immunohistochemistry staining [21]. In addition to the diagnostic and therapeutic advances at the benchside, the recent imaging advances at the bedside have enabled the targeting of individual lesions as an alternative or add-on to the application of pharmaceuticals.

To characterize individual lesions in clinical practice, often, semantic features that, for example, describe the shape, e.g., “spiculation”, the location, or the appearance/texture (“ground glass component”) are used. These markers have been shown to have a certain clinical relevance, but they do not cover the entire range of facets in tumoral biology. In comparison, quantitative imaging biomarkers can help gain objective and quantifiable information, such as radiomics or deep learning features. Radiomics has, for example, been shown to expose a dimension of data beyond a semantic visual characterization that can add valuable information to the predictive power of semantic features [22].

Until now, in contrast to the individual lesion characterization methodologies in molecular oncology and individual targeted therapies that have already been introduced to clinical research and practice, the imaging-based assessment of molecular characteristics has not been applied to translate these developments based on quantitative imaging biomarkers, such as aggressiveness or lesion mutational status, into optimized, personalized treatment. Therefore, quantitative imaging parameters should be deployed and designed with existing approaches already used in molecular oncology in mind in order to achieve a synergistic benefit for treatment planning.

### 1.4. Molecular Tumor Diagnostics for Personalized Therapy Stratification

Methodologies that enable molecular characterization by quantitative analyses are summarized as omics methodologies. Important examples of these techniques include genomics, transcriptomics, proteomics, and epigenomics. Recent advances in technologies such as whole-genome sequencing have enabled researchers to gain unprecedented insights into tumoral characteristics and behavior. The omics methodologies have been applied to a broad range of diseases. As some examples, survival in lung adenocarcinoma can be predicted by methylation signature [23], survival in patients with cancer of unknown primary can be improved with the use of whole-genome sequencing for therapy selection [24], and relapse in non-small lung cell cancer can be predicted by analyses of oncogenic mutations [25]. Via the combination of the different omics analyses, multi-omics analyses were established, making it possible to measure the different levels (e.g., mutational (genomics) and/or DNA methylation (epigenomics)) at which a pathway is affected. Using multi-omics analyses, for example, survival and drug response can be predicted in breast cancer by combining different molecular biomarkers, such as copy number variation or gene expression [26], and survival can be predicted in liver cancer [27]. Furthermore, the analysis of tumor mutational burden can be used to predict response to immunotherapies and thus can for therapeutic stratification [28,29]. Molecular genetic tumor profiling as a prerequisite for the administration of targeted therapeutics is defined as companion diagnostics and represents an integrated part of oncologic diagnostics.

## 2. Techniques for Lesion Analysis Using Quantitative Imaging Biomarkers

Starting from the deciphering of novel imaging markers using molecular oncology methodology, corresponding imaging biomarkers would help to solve challenges in oncologic imaging characterization, e.g., they can support the identification of aggressiveness and response. This can be employed as decision support for selecting lesions for locoregional therapy. In addition, it can support the identification of lesional etiology in cancers of unknown or multiple primaries. Furthermore, it could also be used as a measure of aggressiveness or in the situation of mixed response. Lesion characterization could be utilized for comparative analysis of primary and metastasis to identify driver mutations via imaging. Imaging biomarker analysis can be applied to a large scale of images—from computed tomography or MRI to ultrasound and nuclear imaging. Novel personalized oncologic imaging biomarkers in comparison with traditional imaging biomarkers are summarized in Figure 1.

To tackle the challenges of lesion-specific characterization, multiple methodologies of quantitative imaging biomarkers are feasible. The two most prominent for the identification of imaging biomarkers are classical radiomics features and deep learning.

### 2.1. Classical Radiomics Analysis Workflow and Texture Visualization

Radiomics is a term for the extraction of mostly non-human readable quantitative imaging data from radiological images [30]. The methodology has been a subject of clinical research for several years and is at the threshold of clinical application in certain cases. One of the main reasons for the incomplete translation of research findings in this field into practice is the partial reproducibility of radiomics features [31]. Radiomics features vary depending on different factors, such as ROI size [32], segmentation procedures [33], choice of scanner [34], scanning parameters (including tube current, slice thickness, and reconstruction algorithms [35]), and extraction software [36]. There is also a randomness factor in the analysis, as even coffee break test–retest analyses may not always agree [37]. To reduce the differences between the various extraction softwares and increase reproducibility, the Image Biomarker Standardization Initiative (IBSI) was founded [38]. In total, the IBSI defined 174 shape, first-order (histogram/intensity-based), and second-order (texture-based) features.

Although, methodologically, there are certain limitations, radiomics has been proven to be powerful in a diverse range of use cases in research. Often, radiomics is used in oncologic diseases, such as lung cancer [39], colorectal cancer [40], malignant lymphoma [41], prostate cancer [42], hepatocellular carcinoma [43], renal cell carcinoma [44], and breast cancer [45]. Apart from oncologic diseases, radiomics has been used in neurological diseases, such as Alzheimer’s [46], urologic diseases (e.g., kidney stones [47]), pulmonary diseases (e.g., idiopathic pulmonary fibrosis [48]), coronary plaques [49], and meniscus injuries [50].

The radiomics workflow consists of a step-by-step approach: first, regions of interest (ROI) need to be segmented automatically, semi-automatically, or manually. For automated segmentation, preferably, U-Net architectures are used [51]. Then, an extraction algorithm will be applied to calculate certain predefined quantitative features for this region of interest. A commonly used python package for feature extraction is pyradiomics [52]. Radiomics feature algorithms in pyradiomics are based on the IBSI definition containing shape, first order, and second-order features. In MRI studies, normalization is applied to the images to substitute the missing standardized scale. Different filters, such as Laplacian of Gaussian or wavelet filters, can be applied in the extraction process to accentuate distinctive image characteristics, such as edges [53]. As radiomics feature values can be highly redundant, feature selection methods, such as L1 (LASSO) regularization, Pearson correlation coefficient threshold, and random forest-based permutation feature importance calculation, can be applied to select a final feature set out of all features.

The resulting feature values can be used to train traditional machine learning algorithms, such as a random forest or XG Boost classifier [54,55], correlate them with clinical and genetic parameters, and statistically evaluate them by regression analyses or other classifiers. The radiomics workflow is summarized in Figure 2a.

Radiomics features can be visualized by breaking down the ROIs into smaller parts, providing a basis for a better understanding of tumor texture. For example, within a lesion, specific, particularly active areas can be identified and biopsied. Radiomics feature mapping could facilitate the clinical integration of radiomics information for improved characterization of cancer. Radiomics feature activation maps can also support the exploration of tumor biology in research and aid the delineation of specific tumoral regions that are important in disease progression [56].

### 2.2. Deep Learning Analysis Workflow

Deep learning architectures are characterized by analyzing large numbers of data, especially images, using neural networks with multiple layers. Due to better accessibility via open-source python packages for deep learning, such as Tensorflow and Pytorch, deep learning is increasingly used in biomedical research. From early technical developments of neuron-like circuits by McCullogh and Pitts in 1943 [57] to the implementation of backpropagation and multi-layered convolutional neural networks, many fundamental and groundbreaking developments have been made.

To train a model, unsupervised and (semi-) supervised training approaches can be used. In contrast to radiomics, deep learning has the advantage that it does not require prior segmentation, as the network automatically learns the ROIs/key areas. For supervised training approaches, the input data are labeled prior to being presented to the model. The so-called ground truth, on the basis of which the data are labeled, is ideally defined by the traditional gold standard for diagnosis (histology, genetic markers) or a consensus voting of multiple physicians. The input data for supervised approaches are mostly split into train, test, and validation sets, and performance is evaluated at the test set after the model has been trained using the training and validation set.

Compared to traditional radiomics-based machine learning models, convolutional neural networks have a more generalist approach and are not bound to specific predefined quantitative parameters. Therefore, they may generalize better in specific scenarios. In relation to the computational requirements of radiomics trained machine learning algorithms, deep learning on a large scale can only be performed on supercomputers with powerful GPUs. Therefore, they may generalize better in specific scenarios. A general problem in many research studies is training models in single-scanner, single-center settings. Algorithms need to be trained with heterogeneous datasets that reflect real-world settings as a condition for clinical applications [58]. In comparison to traditional machine learning, deep learning needs a vast number of data to build stable and generalizable models. Such quantities of data are often not available for rare diseases. In the case of a lack of images for training, various techniques can be applied. In particular, augmentation can be used, in which images are rotated, flipped, or otherwise manipulated and added, to increase the number of training data [59]. In addition, transfer learning, where pretrained models are trained on other datasets and fine-tuned with their own data, can be applied [60,61,62]. A typical workflow for the application of CNNs for medical imaging analysis is shown in Figure 2b.

An important limitation of all deep learning models is the risk of biased labeling of data that could, for example, lead to race- and gender-based discrimination [63]. This danger and limitation in artificial intelligence (AI) needs to be addressed in study planning and is being considered increasingly in the research on “ethical AI”, which is concerned with any kind of ethical issues raised by AI [64,65].

In the future, self-attention-based AI techniques (transformer networks), which are currently applied in natural language processing, may help enhance the performance of neural networks in image analysis even further [66,67].

Examples of the quantitative imaging biomarkers presented in this section are shown in Figure 3.

## 3. Application and Evidence for Novel Imaging Biomarkers for Improved Lesion Characterization and Prognosis

A broad range of applications of imaging biomarkers for lesion differentiation in various diseases is feasible. For example, radiomics has already been used to distinguish small hepatocellular carcinomas from focal nodular hyperplasia liver lesions [68], to classify the primary of spinal metastases [69], distinguish the primaries of metastases in lung lesions [70], distinguish benign from malignant lesions in the parotid gland [71], and identify subtypes of renal cell carcinoma [44].

In a complementary way, deep learning has already been used for the differential diagnosis of liver lesions [72,73], the detection of malignant lung nodules [74,75], the detection and classification of breast masses [76], and the assessment of lymph node metastases in women with breast cancer [77]. The methodology described before can be applied to multiple lesional imaging criteria.

Generally speaking, four levels of imaging biomarker usage for lesion differentiation and intralesional characterization can be delineated. First, the lesion’s dignity and etiology can be determined. Second, tumoral heterogeneity can be assessed by imaging biomarkers. Third, a radiogenomic characterization can help to identify lesion aggressiveness and response. Finally, imaging biomarkers can support targeting for biopsy and therapy. An overview of the criteria and methodology for imaging biomarkers is shown in Figure 4.

### 3.1. Lesion Dignity and Etiology Assessment

The differentiation of benign and malignant lesions and classification of them as primary or metastatic are critical clinical procedures in therapy planning. While visual criteria may guide lesion classification, the first evidence has been provided that novel imaging biomarkers can help in this regard and increase diagnostic accuracy.

In breast MRI, the application of a radiomics approach was able to perform lesion assessment and benign/malignant classification with similar accuracy to radiologists [78]. In identifying malignant lesions via mammography, radiomics classifiers could achieve an area under the curve of 0.90 in the test set [79]. Additionally, radiomics characteristics could be employed to distinguish benign and malignant peripheral nerve sheath tumors in neurofibromatosis type 1 with an area under the curve (AUC) of up to 0.94 [80].

### 3.2. Assessment of Tumoral Heterogeneity

When dealing with lesion characterization, the concept of tumoral heterogeneity, in particular, may be necessary to analyze. Heterogeneity can be subdivided into intra- and intertumoral heterogeneity. Intratumoral heterogeneity describes the heterogeneity inside a lesion, while intertumoral heterogeneity describes heterogeneity between different lesions.

In accordance with the concept of intertumoral heterogeneity, research studies have been able to show how the biology can differ from lesion to lesion. Interlesional molecular genetic heterogeneity has been detected in several cancers. In a disseminated tumor, in addition to the destructive growth of metastases into vital organs and vessels, there is also the issue of molecular genetic differentiation of metastatic lesions compared to the primary tumor, so that lesions with different degrees of therapy responsiveness and aggressiveness may coexist in the same patient at the same time. For example, phylogenetic analysis in ovarian cancer showed a high degree of heterogeneity over the course of time and in different lesions [81]. A multi-region analysis study by Gerlinger et al. was able to prove intratumoral heterogeneity and tumor evolution in primary renal carcinomas [82]. Heterogeneity has an effect on tumoral behavior, as it can change the degree of therapy response and lesion aggressiveness [83]. A quantitative assessment of heterogeneity could support decision making in lesion targeting for biopsy and locoregional interventional treatment. The assessment of interlesional heterogeneity based on imaging has not yet been studied comprehensively. First analyses, for example, by the utilization of radiomics parameters for unsupervised machine learning algorithms such as clustering, have been shown to be able to identify lesional subtypes and therefore provide a better understanding of imaging-based tumoral heterogeneity. For example, radiomics signatures of colorectal liver metastases can be clustered and subgroups of liver metastases correlated with clinical parameters can be identified [84]. PET analyses could also be used to identify tumoral heterogeneity by combining the information for multiple targets [85]. It is conceivable that quantifiable information about heterogeneity may be used as a theranostic tool to predict response and relapse.

### 3.3. Assessment of Aggressiveness and Response

The concept of tumor aggressiveness plays a vital role in therapy planning, as fast- growing and destructive tumors require a different intensity of therapy compared to slow-growing and less destructive tumors. The level of aggressiveness could be used to predict outcomes. Different works with volumetric assessments have shown in follicular lymphoma [86], colorectal cancer liver metastases [87], endometrial carcinomas [88], and peripheral T-cell carcinoma [89] that larger tumors tend to be more aggressive and have an influence on outcomes. However, volumetric assessment is of limited validity. Besides traditional analyses, it was demonstrated in patients with colorectal cancer that readily available biomarkers, such as CT attenuation/Hounsfield intensity values, could also be predictors of survival [20]. In line with these results, it was possible to use radiomics for aggressiveness prediction in papillary thyroid cancer using multiparametric MRI [90], in prostate cancer using bi-parametric MRI [91], and in diffuse lower-grade gliomas using diffusion- and perfusion-weighted sequences [92]. For prostate cancer, it was shown that differentiation and aggressiveness assessment using a radiomics-based model outperforms the traditional PI-RADS visual assessment. Chen et al. assessed the radiomics signature from a conventional and diffusion-weighted MRI. In their study, a radiomics trained logistic regression classifier (AUC = 0.985, 0.982, 0.999 (T2WI, ADC, T2WI, and ADC)) outperformed the PI-RADS V2 (AUC = 0.867) in diagnosing prostate cancer [93].

In modern therapy planning, the determination of individual genetic profiles is becoming increasingly important, if not almost essential. Linking mutation status to imaging phenotypes using quantitative imaging biomarkers can help to guide treatment decisions and improve survival prognosis. With computed tomography of lung adenocarcinoma, it is already possible to link mutational status (EGFR, KRAS, and ALK) with imaging-assessed characteristics, such as lymphangitis, pleural effusion, or lung metastases [94].

The discipline of linking genetics with imaging is called radiogenomics. Radiogenomic analyses have already been very successful in identifying various mutations [95]. Some examples of studies for the linking of radiomics and genetics are the detection of BRCA gene status in epithelial ovarian cancer [96], p53 and PD-L1 status in pancreatic cancer [97], the detection of p53 and IDH mutations in gliomas [98,99], the detection of EGFR status in brain metastases of lung adenocarcinoma [100], and KRAS status in rectal cancer [101]. Similar to radiogenomics, imaging-based deep learning and genetics have been linked, too. Thus, it was possible to detect IDH mutation status and PTEN mutation status in gliomas [102,103], KRAS status in patients with colorectal carcinoma [104], and EGFR status in lung adenocarcinoma [105]. Image-based recognition of genetic information can be highly relevant as a theranostic agent. For example, it has been demonstrated that EGFR mutations can change the response for EGFR tyrosine kinase inhibitors [106]. Thus, radiogenomics can serve the molecular classification of certain tumor subtypes, enabling rapid and precise diagnostic decisions. In case of disease progression or recurrence, radiogenomic approaches can identify aggressive and progredient lesions to guide therapeutic decisions in combination with molecular genetic tumor profiling either from biopsy or blood.

Furthermore, radiomics can be used to identify and quantify lesion imaging signatures and correlate them with response, aggressiveness, or growth patterns. First analyses have been able to identify subtypes in colorectal cancer liver metastases [84] and identify volumetric habitats in glioblastomas [107].

### 3.4. Targeting for Biopsy and Therapy

The studies presented here on tumoral heterogeneity and individual lesion biology suggest that biopsies of single lesions at a single time point during a patient’s journey cannot represent the full range of genetic diversity. Individual molecular biological analysis studies have shown how tumor biology differs over time and between different lesions [108]. Thus, smaller but more aggressive lesions might be missed and larger but less aggressive lesions biopsied instead. Furthermore, lesions could be fully missed if non-guided biopsies are performed. The usage of quantitative imaging biomarkers could improve targeting. This is notably important in the case of prostate carcinoma, where the spatial conditions mean that a solely MRI-guided biopsy does not always include clinically significant prostate cancer [109]. For example, combining prostate-specific antigen density and MRI for prostate biopsy planning allows the optimization of biopsy planning [110]. Interlenghi et al. were even able to predict the BI-RADS category for suspicious breast masses using ultrasound radiomics and training-modified random forest classifiers, support vector machines, and k-nearest neighbor classifiers [111]. Further developments could even lead to radiomics replacing biopsy and facilitate therapeutic decisions by predicting BI-RADS category by imaging.

## 4. Future Perspectives

The applications of quantitative imaging biomarkers presented in this paper are still in their infancy compared to other methods. Despite trends in research, for everyday clinical practice, deep learning or radiomics-based models are used sporadically, if at all. In the first place, the modern multi-omics technologies developed in molecular biology have made it possible to generate this new dimension of individual lesion characterization based on imaging. Without molecular biological multi-omics analyses, no ground truth can be defined and models cannot be trained properly. To enhance the progress in individual characterization, large datasets of multi-omics analyses need to be obtained to develop image-based markers. This could, for example, include a revival of research autopsies, which can help to understand tumor evolution [112]. Even though there are a large number of studies on specific questions using quantitative imaging biomarkers, there is a lack of more general methods that are, for example, scanner-independent.

In the future, a strong collaboration between laboratory medicine, pathology, and radiology will be essential to provide the best possible personalized therapy for patients with oncologic diseases. Models could be trained with a combination of imaging biomarkers and other non-invasively determined biomarkers, such as liquid profiling analyses. Liquid profiling describes the investigation of tumor genetics by blood analysis to determine circulating tumor DNA. Due to the lower hurdle of such a liquid biopsy compared to a traditional biopsy, genetic analyses can be performed at multiple time points [113]. This methodology has enormous potential and it could be employed as a personalized tumor marker, using patient- and tumor-specific alterations for tracking the genetic tumor evolution in blood. Importantly, the value as part of an integrative diagnostic approach in combination with radiomics to observe clonal heterogeneity in non-small cell lung cancer has been demonstrated recently [114].

Imaging biomarker analyses could be made part of clinical software (e.g., PACS) to provide integrated data analysis and risk stratification, prediction, and individual lesion characterization. By automatically incorporating them into diagnostic pathways, they could be used as a standardized element in therapy planning and clinical decision making in general.

## 5. Conclusions

Imaging biomarkers have great potential for lesion characterization and can help to foster the progress already made through molecular biological discoveries in a more low-threshold clinical setting. The potential applications, especially for lesion-specific assessment, are considerable, as shown by the many studies presented in this work. In order to improve quality of care and exploit the possibilities of personalized therapy, image-based assessment of tumor behavior is a promising approach for the future.

## Figures and Tables

**Figure 1 cancers-14-03349-f001:**
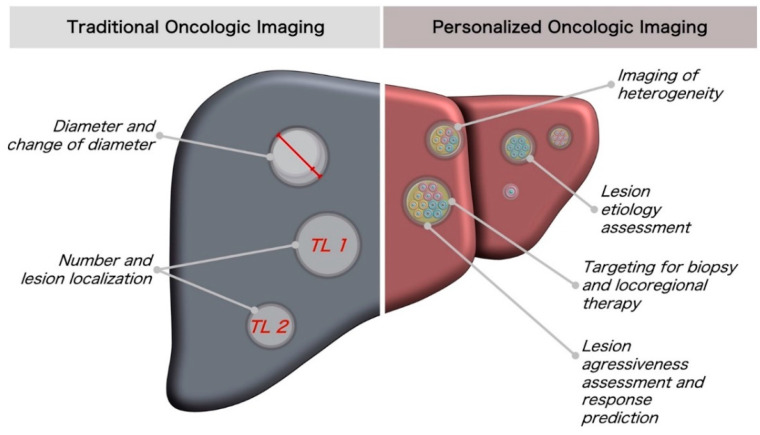
Comparison of traditional oncologic imaging approaches and quantitative imaging biomarkers for better lesion characterization toward personalized oncologic imaging.

**Figure 2 cancers-14-03349-f002:**
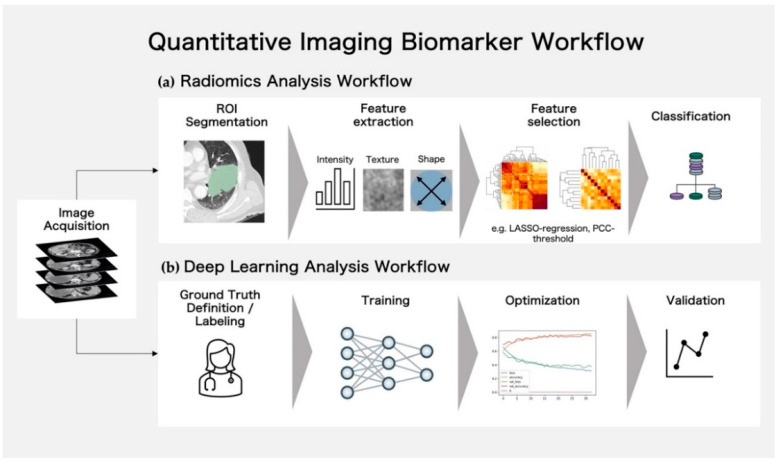
Analysis workflows for quantitative imaging biomarkers. (**a**) Workflow for radiomics analysis: Regions of Interest (ROIs) are segmented manually or automatically, following features of different categories (intensity-, texture-, or shape-based), which are extracted for each ROI. Extracted features are reduced and important features are outlined in the feature selection process. The final selection feature set can be used for classification. (**b**) Workflow for deep learning analysis: for supervised approaches the ground truth needs to be defined/the data need to be labeled. Following this, the neural networks can be trained, the learning process can be optimized by hyperparameter tuning, and the model validated on an external/unseen dataset.

**Figure 3 cancers-14-03349-f003:**
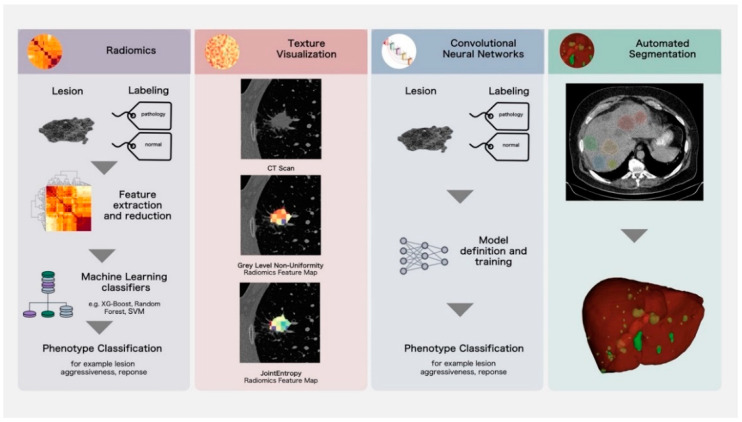
Overview of imaging biomarker methodology.

**Figure 4 cancers-14-03349-f004:**
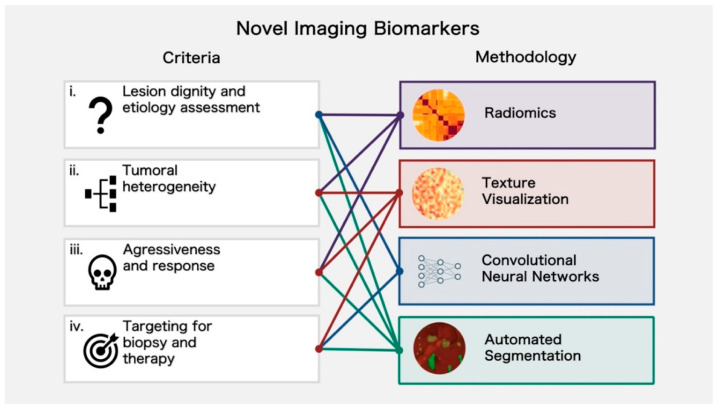
Criteria and methodology for novel imaging biomarkers. To solve various challenges, different methodologies can be used. This figure shows ideally applicable methods for (i) lesion dignity and etiology assessment, (ii) tumoral heterogeneity, (iii) aggressiveness and response, and (iv) targeting for biopsy and therapy.

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
