# Peer review of "The Potential and Emerging Role of Quantitative Imaging Biomarkers for Cancer Characterization"

_cancers, 2022, doi:10.3390/cancers14143349_

Round 1

Reviewer 1 Report

I would like to congratulate the authors for this comprehensive review of literature on a state-of-the-art subject. The authors used pertinent and high quality references in addition to a very well structured and logical exposure of ideas. However, it might be of value to present the clinician with a few details regarding the type of imaging studies amenable for becoming biomarkers, and if there is a need of extraordinary equipments, such as super-computers. Since the subject of imaging biomarkers is of relative novelty, it is important to understand how it will be possible to include these type of assessments in clinical routine in the future.

I find this bringing up to speed of lecturers on the field of Radiomics find very useful, since this direction of personalised medicine is showing great potential for development. 

Author Response

I would like to congratulate the authors for this comprehensive review of literature on a state-of-the-art subject. The authors used pertinent and high quality references in addition to a very well structured and logical exposure of ideas.

We thank the reviewer for this very positive feedback.

However, it might be of value to present the clinician with a few details regarding the type of imaging studies amenable for becoming biomarkers, and if there is a need of extraordinary equipments, such as super-computers. [1] Since the subject of imaging biomarkers is of relative novelty, it is important to understand how it will be possible to include these type of assessments in clinical routine in the future.

We added a paragraph for the imaging studies amenable for biomarker analyses:

Imaging biomarker analysis can be applied to a large scale of images – from Computed Tomography or MRI to ultrasound.

We also added a paragraph for the technical requirements:

In relation to the computational requirements of radiomics trained machine learning algorithms, deep learning on a large scale can only be performed on supercomputers with powerful GPU’s.

I find this bringing up to speed of lecturers on the field of Radiomics find very useful, since this direction of personalised medicine is showing great potential for development.

Reviewer 2 Report

the review is clearly organized and the different section provide the reader with different examples of fields of application for quantitative imaging biomarkers in oncology.

Considering how broad and superficial the scope of this review is compared to the complexity of the subject, it is clear that this mauniscript is not intended for experts or active imaging scientists already working in the field. Therefore, i suggest adding a short paragraph in the introduction section, as well as a mention in the abstract, regarding the purpose of this review and describing your intended audience (eg it could prove most informative for radiologists who are planning to conduct radiomics or machine-learning projects, radiology residents wishing to understand the application of radiomics, oncologists etc).

I would also suggest expanding the description of a few studies mentioned in the text, which represent successful applications of radiomics parameters for the assessment of tumour heterogeneity, aggressiveness and biopsy targeting, most suitable for providing the reader with more concrete evidence of the value of these image-based approaches. Therefore, similarly to how was done with reference 79 in paragraph 3.1, i encourage the authors to provide slightly more in-depth descriptions of the references n°84 in paragraph 3.2, n°93 in paragraoh 3.3 and n°111 in paragraph 3.4. I feel that the discussion of these papers was too vague and uninformative.

Author Response

The review is clearly organized and the different section provide the reader with different examples of fields of application for quantitative imaging biomarkers in oncology.

Thank you very much for this positive feedback.

Considering how broad and superficial the scope of this review is compared to the complexity of the subject, it is clear that this mauniscript is not intended for experts or active imaging scientists already working in the field. Therefore, i suggest adding a short paragraph in the introduction section, as well as a mention in the abstract, regarding the purpose of this review and describing your intended audience (eg it could prove most informative for radiologists who are planning to conduct radiomics or machine-learning projects, radiology residents wishing to understand the application of radiomics, oncologists etc).

We considered your recommendations and revised following sentence in the abstract:

“This review gives a methodological introduction for clinicians interested in the potential of quantitative imaging biomarkers, in particular into radiomics models, texture visualization, convolutional neural networks and automated segmentation.”

I would also suggest expanding the description of a few studies mentioned in the text, which represent successful applications of radiomics parameters for the assessment of tumour heterogeneity, aggressiveness and biopsy targeting, most suitable for providing the reader with more concrete evidence of the value of these image-based approaches. Therefore, similarly to how was done with reference 79 in paragraph 3.1, i encourage the authors to provide slightly more in-depth descriptions of the references n°84 in paragraph 3.2, n°93 in paragraoh 3.3 and n°111 in paragraph 3.4. I feel that the discussion of these papers was too vague and uninformative.

Thank you for this detailed assessment and suggestions. We have taken your recommendations in account and modified the paragraphs 3.1.:

3.2., ref 84: . First analyses, for example, by the utilization of radiomics parameters for unsupervised machine learning algorithms such as clustering, have been shown to be able to identify lesional subtypes and, therefore, provide a better understanding of imaging-based tumoral heterogeneity. In a study by our group radiomics signatures of colorectal liver metastases were clustered and subgroups of liver metastases correlated to clinical parameters were identified [84]. 

3.3., ref 93: For prostate cancer, it was shown that the differentiation and aggressiveness assessment using a radiomics-based model outperforms the traditional PI-RADS visual assessment. Chen et al. assessed the radiomics signature from a conventional and diffusion-weighted MRI. In their study a radiomics trained logistic regression classifier (AUC = 0.985, 0.982, 0.999 (T2WI, ADC, T2WI&ADC)) outperformed the PI-RADS V2 score (AUC = 0.867) in diagnosing prostate cancer [93]. 

3.4. ref 111: Interlenghi et al. were even able to predict the BI-RADS category for suspicious breast masses using ultrasound radiomics and training modified random forest classifiers, support vector machines, and k-nearest neighbor classifiers[111]. Further developments could even lead to radiomics replacing biopsy and facilitate therapeutic decisions by predicting the BI-RADS category by imaging.

Reviewer 3 Report

In their review article 

"The potential and emerging role of quantitative imaging biomarkers for cancer characterization"

the authors describe very well role of imaging biomarkers in cancer imaging. The article is very well written and I liked reading it. The only point I like to bring up is, that "imaging biomarkers" should include moleculare/nuclear medicine imaging which is just mentioned along. If the authors are not willing to extent the article in this direction, what I would prefere, I suggest to change the title accordingly.

Otherwise some minor spell checking is required before publication.

Author Response

We thank the reviewer very much for the positive feedback. We totally agree with the importance of the broad applicability of imaging biomarkers. For this reason we highlighted it.

Imaging biomarker analysis can be applied to a large scale of images – from Computed Tomography or MRI to ultrasound and nuclear imaging. 

Language spell checks were performed and revised.

Reviewer 4 Report

In the manuscript “The potential and emerging role of quantitative imaging biomarkers

for cancer characterization”, authors give an account of quantitative imaging methodologies from the angle of oncology.

The simple summary and the abstract are well-written.

The paper structure is good, broken down into manageable sections, and each section is accompanied by a figure. The content does not contain detectable errors.

However:

The paper is currently far from addressing readers outside the field.

Specifically the manuscript is currently not reader-friendly for readers from other fields. Example:

2.2. Deep Learning Analysis Workflow
Deep learning architectures are characterized by analyzing large amounts of data,
especially images, using neural networks with multiple layers. Due to better accessibility
via packages like Tensorflow and Pytorch, deep learning is increasingly used in biomed-
ical research.

“Packages” is a fine expression, when addressing readers of the same field.

- Not too fine to let other readers, from unrelated fields, guess that this refers to software (open-source, …). It may seem counter-intuitive, but this distracts the reader who has not used, or considered using Pytorch, or Tensorflow.

Otherwise, the main text, although in general well-conceived, cannot, or cannot yet be described as a “must-read”. This despite of the fact that the main text is well-structured, and illustrated with figures. Even though the text lacks major errors, it is difficult to read and needs special attention to the use of sentence building.

The figures represent the main sections of the text. Figures and figure legends are currently not helpful. As they are now, especially figures 2 and 4, are not easy to comprehend without effort. Either the legends need to change, or the figures, to make each figure self-explanatory. One potential solution is to change the figure content and make the illustration more descriptive. The other solution is to change the legends, and make them explain in very short but effective terms, what one sees in the image shown.

Finally, even though the text is up-to-date in terms of content, it needs work to make it flow better. One example are some of the large paragraphs, which contain many different pieces of information, without to allow the reader to retain overview, or without an overarching theme.

For instance, perhaps pieces of text like

The assessment of interlesional heterogeneity based on imaging has not yet been studied comprehensively. First analyses, for example, by the utilization of radiomics parameters for unsupervised machine learning algorithms such as clustering, have been shown to be able to identify lesional subtypes and, therefore, provide a better understanding of imaging- based tumoral heterogeneity [84]. PET analyses could also be used to identify tumoral

heterogeneity by combining the information of multiple targets [85]. It is conceivable that

the use of quantifiable information about heterogeneity as a theranostic may predict response

and relapse.”

Could form a new paragraph.

Another example is the title “3. Evidence for novel imaging biomarkers for improved lesion characterization and prognosis”

This title is not connected to the section that it refers to.

Either use a title similar to “3. Application of novel imaging biomarkers for improved lesion characterization and prognosis”

Or edit the text of the section as evidence. So that the section title will indeed correspond to the content of this section.

Author Response

In the manuscript “The potential and emerging role of quantitative imaging biomarkers for cancer characterization”, authors give an account of quantitative imaging methodologies from the angle of oncology.

The simple summary and the abstract are well-written. The paper structure is good, broken down into  manageable sections, and each section is accompanied by a figure. The content does not contain detectable errors.

We thank the reviewer for this positive feedback.

However:

The paper is currently far from addressing readers outside the field.

Specifically the manuscript is currently not reader-friendly for readers from other fields. Example:

2.2. Deep Learning Analysis Workflow

Deep learning architectures are characterized by analyzing large amounts of data,

especially images, using neural networks with multiple layers. Due to better accessibility

via packages like Tensorflow and Pytorch, deep learning is increasingly used in biomed-

ical research.

“Packages” is a fine expression, when addressing readers of the same field.

- Not too fine to let other readers, from unrelated fields, guess that this refers to software (open-source, …). It may seem counter-intuitive, but this distracts the reader who has not used, or considered using Pytorch, or Tensorflow.

“Deep learning architectures are characterized by analyzing large amounts of data,

especially images, using neural networks with multiple layers. Due to better accessibility

via open source python packages like Tensorflow and Pytorch, deep learning is increasingly used in biomedical research.

Otherwise, the main text, although in general well-conceived, cannot, or cannot yet be described as a “must-read”. This despite of the fact that the main text is well-structured, and illustrated with figures. Even though the text lacks major errors, it is difficult to read and needs special attention to the use of sentence building.[2] 

The figures represent the main sections of the text. Figures and figure legends are currently not helpful. As they are now, especially figures 2 and 4, are not easy to comprehend without effort.[3]  Either the legends need to change, or the figures, to make each figure self-explanatory. One potential solution is to change the figure content and make the illustration more descriptive. The other solution is to change the legends, and make them explain in very short but effective terms, what one sees in the image shown.

We thank the reviewer for this suggestion. We completely agree and have revised the figure descriptions.

Figure 2: Analysis workflows for quantitative imaging biomarkers  (a) Workflow for radiomics analysis: Regions of Interest (ROI) are segmented manually or automatically, following features of different categories (intensity, texture or shape based) are extracted for each ROI. Extracted features are reduced and important features are outlined in the feature selection process. The final selection feature set can be used for classification; (b) Workflow for deep learning analysis: For supervised approaches the ground truth needs to be defined/ the data needs to be labeled. Following, the neural networks can be trained, the learning process can be optimized by hyperparameter tuning and the model validated on an external/unseen dataset.

Figure 4. Criteria and Methodology for Novel Imaging Biomarkers. To solve various challenges different methodologies can be used. This figure shows ideally applyable methods for i. Lesion dignity and etiology assessment, ii. Tumoral heterogeneity, iii. Aggressiveness and response, and iv. Targeting for biopsy and therapy.

Finally, even though the text is up-to-date in terms of content, it needs work to make it flow better. One example are some of the large paragraphs, which contain many different pieces of information, without to allow the reader to retain overview, or without an overarching theme.

For instance, perhaps pieces of text like

“The assessment of interlesional heterogeneity based on imaging has not yet been studied comprehensively. First analyses, for example, by the utilization of radiomics parameters for unsupervised machine learning algorithms such as clustering, have been shown to be able to identify lesional subtypes and, therefore, provide a better understanding of imaging- based tumoral heterogeneity [84]. PET analyses could also be used to identify tumoral heterogeneity by combining the information of multiple targets [85]. It is conceivable that the use of quantifiable information about heterogeneity as a theranostic may predict response and relapse.”

Could form a new paragraph.

Another example is the title “3. Evidence for novel imaging biomarkers for improved lesion characterization and prognosis”

This title is not connected to the section that it refers to.

Either use a title similar to “3. Application of novel imaging biomarkers for improved lesion characterization and prognosis” or edit the text of the section as evidence. So that the section title will indeed correspond to the content of this section.

We have implemented the two specific suggestions and have re-read and optimized the readability.

“3. Application and evidence for novel imaging biomarkers for improved lesion characterization and prognosis”

Reviewer 5 Report

In this review authors summarize the following methods used in quantivitive imaging of neuronal cancer setting particularly on focus of personalized miedicine.

In prinicple this is a very good and well-structure and well-written review and should be published.

Minor comments

Authors should revise intravital imaging for personalized medicine (diagnosis and therapy). We are thinking about microstuructre (i.e. sinusoidal level in HCC). This would be great, since this structure most likely proceed cancer transformation, meaning prognostic values.

Also, applixcability of this imaging on experimental setting (mice vs rats). This will be fundamental to analyse tumour development and progression spatio-temporal in live subject.

How one can couple this imaging procedures with molecular profiling (OMICS).

Is there anyway to use AI or deep learning to identify tumour nodules as early as possible?

Author Response

In this review authors summarize the following methods used in quantitative imaging of neuronal cancer setting particularly on focus of personalized miedicine.

In prinicple this is a very good and well-structure and well-written review and should be published.

 Minor comments

Authors should revise intravital imaging for personalized medicine (diagnosis and therapy).[4]  We are thinking about microstuructre (i.e. sinusoidal level in HCC). This would be great, since this structure most likely proceed cancer transformation, meaning prognostic values. Also, applixcability of this imaging on experimental setting (mice vs rats). This will be fundamental to analyse tumour development and progression spatio-temporal in live subject.

How one can couple this imaging procedures with molecular profiling (OMICS).

Is there anyway to use AI or deep learning to identify tumour nodules as early as possible?

We added more information about the coupling of multi-omics with quantitative imaging biomarkers in this chapter:

In the first place, the modern multi-omics technologies developed in molecular biology have made it possible to generate this new dimension of individual lesion characterization based on imaging. Without multi-omics no ground truth can be defined and the models cannot be trained properly. To enhance the progress in individual characterization, large datasets of multi-omics analyses need to be obtained to develop image-based markers. This could, for example, include a revival of the research autopsies, which can help to understand tumor evolution [112]. Even though there are a large number of studies on specific questions using quantitative imaging biomarkers, there is a lack of more general methods that are, for example, scanner-independent. 

We thank the reviewer for the very positive feedback and refer to the above mentioned changes.

Round 2

Reviewer 4 Report

Still far from ideal, but it can be published.